# Exploring the influence of COVID-19 stress on mental health among international undergraduate and graduate students: A mixed-methods approach

Chinyere N. Reid[1], Jason W. Beckstead[1], Abraham A. Salinas-Miranda[1,2]*

1 University of South Florida, College of Public Health, Community Health Sciences Department, Tampa, Florida, United States of America, 2 University of South Florida, College of Public Health, Community Health Sciences Department, James and Jennifer Harrell Center for the Study of Family Violence, Tampa, Florida, United States of America

* asalinas@usf.edu

## Abstract

International college students in the United States are at increased risk of developing mental health disorders and are less likely to seek mental health services. However, little is known about the effects of stress during the COVID-19 pandemic on the mental health of international undergraduate students compared to their graduate counterparts studying in the US. This study examined the associations between COVID-19 stress, anxiety, and depression and whether education level moderates these associations in international undergraduate and graduate students. A cross-sectional online survey containing psychometric scales and open-ended questions was completed by 219 international undergraduate and graduate students. Validated psychometric scales used included the Depression, Anxiety, and Stress Scale (DASS-21), COVID-19 Stress Scale (CSS), and Perceived Stress Scale 4 (PSS-4). Path analysis was used to assess whether education level moderated the relationships between COVID-19 stress and anxiety and depression. Applied thematic analysis was conducted to qualitatively determine COVID-19-related stressors affecting students' mental health. We found that COVID-19 stress was significantly associated with students' anxiety and depression, and education level moderated the relationship between COVID-19 stress and anxiety, but not depression. Major themes of COVID-19-related stressors affecting mental health included academic performance, financial difficulties, travel constraints, social isolation, and grief. Findings highlighted the influence of stress during the COVID-19 pandemic on the mental health of international students studying in the US. Graduates and undergraduates experienced stress differently, particularly related to anxiety. Additionally, pandemic-related stressors were multifaceted in nature. Adequate mental health interventions and support tailored for graduates and undergraduates are needed to address students affected during global crises.

**Data availability statement:** All relevant data are within the manuscript and its Supporting information files (ZIP file provided as Supporting information).

**Funding:** The author(s) received no specific funding for this work.

**Competing interests:** The authors have declared that no competing interests exist.

## Introduction

The COVID-19 pandemic exerted a negative impact on health and socioeconomic indicators globally [1]. Numerous public health measures were enacted including lockdowns, stay-at-home orders, and school closures. Educational systems were also affected [2]. Many college students were displaced from classes, jobs, and on-campus housing, having to relocate elsewhere mid-semester. There was a heightened fear of COVID-19 illness and death, social isolation, financial uncertainty, and academic performance [3,4], which could have contributed to the development of anxiety or depression among college students [5–10].

Adolescents and young adults, including college students, are a vulnerable group for mental health disorders. Nearly 75% of all lifetime mental disorders begin by young adulthood [11]. The global prevalence of anxiety and depression in undergraduate and graduate students is estimated to be 39.0% and 33.6%, respectively, with even higher rates reported since the start of the pandemic [12]. Pre-pandemic, graduate students were found to be more than six times likely to experience anxiety and depression compared to the general population [13].

International students were considered a vulnerable college population disproportionately affected by the COVID-19 pandemic [14]. International college students studying in the United States (US) are at an even higher risk for mental health issues compared to domestic students because of greater number of stressors such as academic stress, acculturative stress, discrimination, loneliness, among other factors have been shown to increase their risk of anxiety and depression [15–18]. Once mitigation restrictions began, many international students were unable to remain in on-campus housing, work on-campus jobs or return to their home countries due to visa and re-entry restrictions into the US. Challenges with remote learning, lack of social support, fear of COVID-19 exposure, stigmatization, and discrimination were also reported [14,19].

Mental health issues affect students' concentration, motivation, and social interactions thereby hindering student success [4]. Studies have found an increase in mental health issues such as stress, anxiety, and depression reported by both US undergraduate and graduate students during the pandemic [20,21], with undergraduate students reporting poorer mental health than graduate students [7,22]. Although some studies have examined the mental health of international students studying at US universities during the COVID-19 pandemic [5,23–25], there is a dearth of research examining the differences in COVID-19 stress and its impact on mental health between undergraduate- and graduate-level international students. Therefore, we conducted this study to characterize the relationships between COVID-19 stress and mental health issues from the perspective of international students, pursuing the following specific objectives: (1) assess if there were differences in the relationship between COVID-19 stress and anxiety among undergraduate and graduate international students (quantitative), (2) assess if there were differences in the relationship between COVID-19 stress and depression among undergraduate and graduate international students (quantitative), and (3) identify and understand the context of additional stressors that impacted the lives of international students' at a large US university during the COVID-19 pandemic (qualitative). We hypothesized that

COVID-19 stress was associated with an increase in anxiety, and the association between COVID-19 stress and anxiety would be stronger among undergraduate international students compared to graduate international students. We also hypothesized that COVID-19 stress was associated with an increase in depression, and the association between COVID-19 stress and depression would be stronger among undergraduate international students compared to graduate international students (Fig 1).

## Materials and methods

### Study design

A convergent parallel study design was conducted using a web-based survey (Fig 2). Thus, we concurrently conducted the quantitative questions and qualitative questions in the same phase of the research, weighed the methods equally, analyzed them independently, and interpreted results together [26].

### Setting

The study was conducted at a large public university in Florida, in collaboration with the university's international student services. Data collection occurred from October 1–25, 2020. The survey was anonymous, designed, and delivered with the web-based software Qualtrics [27].

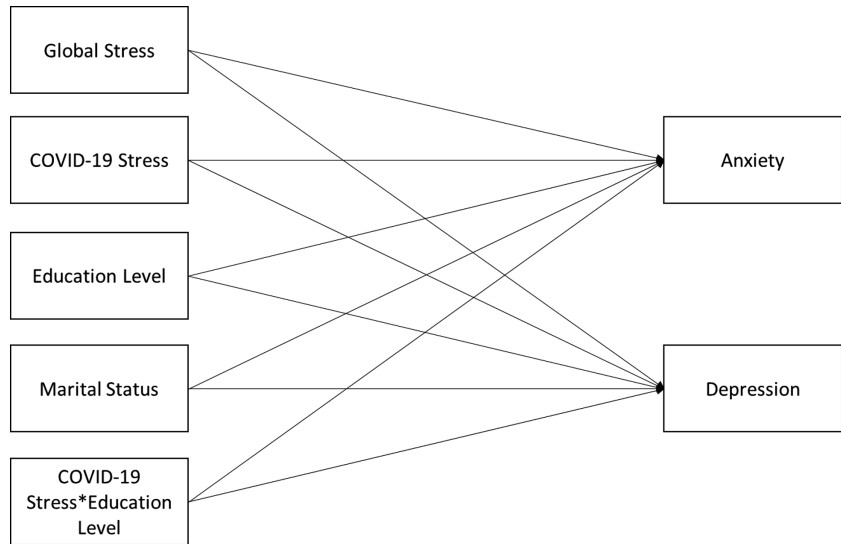

**Fig 1. Path model proposed for the study.**

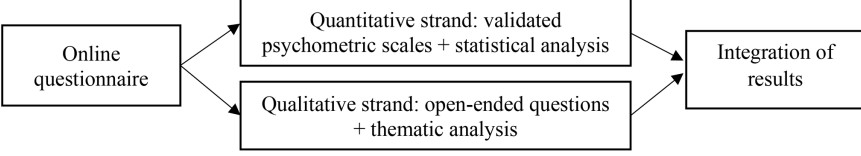

**Fig 2. Convergent parallel design.**

## Participants

A total of 219 participants were included in the study, of which 219 completed all closed-ended questions and 115 gave a response to the open-ended question. Inclusion criteria were self-identified international students, enrolled full time, and aged 18 years or older. Domestic, part-time, and younger students were excluded from the study. Voluntary response sampling was used with a digital flyer and online email to invite volunteers to participate in the study. Email invitations were distributed through the international student listserv at a large public university in Florida and the university's international student services social media sites. Participants provided informed consent electronically at the beginning of the survey.

## Quantitative measures

The quantitative survey consisted of 52 closed-ended questions with validated psychometric scales to assess depression, anxiety, COVID-19 stress, global perceived stress, and demographics.

*Dependent variables.* The outcome variables, depression and anxiety, were each measured using the Depression, Anxiety and Stress Scale (DASS-21) [28]. This scale consists of three scales measuring depression, anxiety, and stress, and each scale consists of seven items scored on a four-point Likert scale. Items were summed into a cumulative numerical score, which was used in the analysis. Higher scores indicated higher levels of the measured outcome. For this study, we used only the depression and the anxiety sub-scales. The depression scale assessed hopelessness, dysphoria, self-deprecation, lack of interest or involvement, devaluation of life, anhedonia and inertia [28]. In our study, the Cronbach's α of the depression scale was 0.94. The anxiety scale assessed situational anxiety, skeletal muscle effects, autonomic arousal, and the subjective experience of anxious affect [28]. In our study, the Cronbach's α of the anxiety scale was 0.85.

*Independent variables.* The COVID-Stress Scales (CSS) were used to measure students' perception of stress related to the COVID-19 pandemic within the last seven days [29]. It is a 36-item scale, which includes five subscales: (1) Danger and contamination fears, (2) fears about economic consequences, (3) xenophobia, (4) compulsive checking and reassurance seeking, and (5) traumatic stress symptoms about COVID-19. The xenophobia subscale was not used in this study because of conceptual mismatch. The subscale items assess xenophobic attitudes towards foreigners (e.g., "*I am worried that foreigners are spreading the virus in my country*"), which was inappropriate for international students in the US who were more likely to have experienced xenophobia rather than displayed it. Therefore, students' experiences of discrimination was not directly measured. To maintain validity, we did not modify the scale. Each item of the CSS was measured on a five-point Likert scale. Items were summated into a cumulative numerical score, which was used in the analysis. Higher total scores indicated higher levels of COVID-19 stress. In our study, the Cronbach's α of the modified-CSS (i.e., excluding xenophobia subscale) was 0.95. Item-total correlations and subscale Cronbach's α for the modified-CSS are shown in S1 Table.

As a measure of global stress, we used the Perceived Stress Scale 4 (PSS-4), which is a 4-item version of the original 10-item scale and was used to assess student's perceived global stress within the last month [30]. The PSS-4 has been found to be highly reliable across cultures [31]. Each item was measured on a five-point Likert scale and summed in a total PSS score, which was used in the analysis. Higher total scores indicated higher levels of perceived global stress. In our study, the Cronbach's α for this scale was 0.71.

Sociodemographic variables used in this analysis included age, gender, education level, and marital status. Age was measured ordinally (18–24 years, 25–30 years, 31–35 years, and 35 + years) but was treated as an ordinal approximation of age as a continuous variable in the analysis. Gender was measured as dichotomous (males, females). Education level was measured as dichotomous (undergraduate, coded as 0 vs. graduate, coded as 1). Marital status was measured as dichotomous (single, married).

## Qualitative measures

We included a single open-ended question asking students *"Has there been any other way that the COVID-19 pandemic has impacted your academics, health, and life?"* to get a greater understanding of the context and factors deemed as stressors during the pandemic.

## Sample adequacy

We invited a listserv of approximately 5000 international students. Our survey was initially completed by 223 students only. After four surveys were excluded due to missing data for all closed-ended questions, our final study sample was 219 participants. This sample was considered adequate for quantitative analysis based on the rule of thumb of a minimum sample size of 200 for path analysis studies [32,33]. Also, we conducted post hoc analysis based on the size of the population (N = 5000 and 95% confidence intervals, using the online Qualtrics margin of error calculator [34]. Our sampling error with our attained sample size was approximately 6.5%, which is within the acceptable range (4–8%) [33].

From a qualitative standpoint, the sample was purposive since all participants were international students during the COVID-19 pandemic.

## Statistical analysis

***Descriptive statistics.*** The CSS is scored on a much higher range than the scores for the other study variables. To make student scores more comparable across variables and improve interpretability of path coefficients, we scaled scores by dividing by 10, which did not affect reliability, significance, or the relative magnitude of model estimates. Frequencies and percentages were calculated for demographic variables (age, gender, education level, and marital status) and medians and interquartile ranges were calculated for COVID-19 stress, global stress, anxiety, and depression variables. Data for four participants were excluded (listwise) from the study due to missing survey item responses, which accounted for only 1.8% of the data. The low proportion of missing data and Little's MCAR test (p = 0.08) indicating missingness was completely at random, supported the appropriateness of using listwise deletion.

***Bivariate analysis and path analysis.*** Pearson's product moment correlation was used to assess the relationship between study variables, including point biserial correlation when one variable was binary and the other continuous. Path analysis using Multiple Likelihood Estimation (MLE) in Mplus 8 [35], was used to assess direct effects and moderation within our hypothesized model with interrelated outcomes. Compared to standard multiple regression which uses ordinary least squares, path analysis (MLE) can estimate complex relationships simultaneously, produce more robust results, and better handle non-normal data [36]. Our assessment of the data's skewness, kurtosis, and histogram plots supported the use of MLE. Endogenous (dependent) variables used in this study were anxiety and depression. Exogenous (independent) variables used were COVID-19 stress and perceived global stress. We assessed the bivariate relationship between demographic variables (age, gender, marital status, education level) to anxiety and depression. Education level and marital status had a significant correlation to anxiety and depression, respectively, and were therefore entered into the model as exogenous variables. The COVID-19 stress variable was mean-centered before calculating the interaction term between COVID-19 stress and education level (moderator), as recommended [37]. Based on significant correlations from bivariate analysis, anxiety was modeled on COVID-19 stress, education level, and the interaction term, and depression was modeled on COVID-19 stress, education level, marital status, and the interaction term. Marital status was not modeled on anxiety because they were not significantly correlated.

We used standardized regression weights (β) to express the magnitude, direction, and significance of relationships between variables in the path model. The overall model fit was assessed to be acceptable based on model fit indices: $\chi^2$ test value is close to zero, $\chi^2$ test p value > 0.05 [38], root mean square error of approximation (RMSEA) ≤ 0.05 [39], comparative fit index (CFI) ≥ 0.95 [40], and standardized root mean square residual (SRMR) ≤ 0.05 [41].

## Qualitative analysis

Responses to the single open-ended question were analyzed qualitatively. Specifically, textual responses were analyzed using an applied thematic analysis approach [42] as follows: 1) familiarization with the data, (2) initial code generation, (3) iterative refinement of the codebook over two cycles of coding, (4) reviewing and refining themes. This process was managed with Microsoft Excel software using its color coding and sorting features [43] by a single coder—an international graduate student researcher. To reduce interpretive bias, peer debriefing involving regular discussions with another researcher was used to critically review coding decisions, resolve any coding discrepancies, and refine interpretation of the data. Data saturation was achieved when no new themes emerged from all participant responses. Student quotes were used to represent themes (confirmability). A narrative contiguous approach was utilized for triangulation [44], which consists of reporting results of the quantitative strand followed by results of the qualitative strand in different subsections of the report. Additionally, a joint display was created to triangulate quantitative and qualitative findings.

## Ethical considerations

This study was determined to be exempt by the University of South Florida Institutional Review Board.

## Results

### Participants characteristics

Student characteristics and psychological variables are described by education level in Table 1. The study population consisted of 108 (49%) undergraduate students and 111 (51%) graduate students. For both undergraduate- and graduate-level, most students were aged 18–24 years, female, and single. Students reported originating from the following geographic regions: South Asia (26.9%), East or Southeast Asia (20.1%), South America (16.4%), Caribbean (12.8%), Europe

Table 1. Participant characteristics and median (IQR) of psychological variables used in the model by education level.

| Characteristic | Undergraduates, n (%) | Graduates, n (%) |
|---|---|---|
| Total (n = 219) | 108 (49.3) | 111 (50.7) |
| Age, years | | |
| 18-24 | 96 (88.9) | 42 (37.8) |
| 25-30 | 11 (10.2) | 35 (31.5) |
| 31-35 | 1 (0.9) | 21 (19.0) |
| 35+ | 0 (0.0) | 13 (11.7) |
| Sex | | |
| Male | 29 (26.9) | 55 (49.5) |
| Female | 79 (73.1) | 56 (50.5) |
| Marital Status | | |
| Single | 103 (95.4) | 93 (83.8) |
| Married | 5 (4.6) | 18 (16.2) |
| **Variable** | **Median (IQR)** | **Median (IQR)** |
| Global Stress | 9.0 (7.5–11.0) | 8.0 (6.0–10.0) |
| COVID-19 Stress | 4.5 (2.7–6.5) | 3.9 (2.5–5.6) |
| Anxiety | 4.0 (1.0–7.0) | 2.0 (0.0–5.0) |
| Depression | 5.5 (1.5–10.0) | 3.0 (1.0–7.0) |

IQR – Interquartile Range.

(11.9%), Africa (4.1%), West Asia (3.7%), Central America (2.7%), and North America—Canada (1.4%). The correlation coefficients of study variables are shown in Table 2. COVID-19 stress was significantly correlated with global stress, but not very highly (r = 0.28, p < .01). COVID-19 stress was significantly correlated with anxiety (r = 0.45, p < .01) and depression (r = 0.34, p < .01).

## COVID-19 stress, anxiety, and depression among international students

*Assessment of model fit.* The initial saturated model (Fig 1) had R-squares of 0.353 and 0.376 for anxiety and depression, respectively. The path of marital status to anxiety was nonsignificant and so removed, and the model was re-estimated. The fit for the revised model was considered a very good fit based on meeting model fit criteria: $\chi^2$ (1) = 0.35, p = 0.56; SMR = 0.009; RMSEA = 0.00 (90% CI:0.00, 0.15); and CFI = 1. The R-squares for the revised model were 0.352 for anxiety and 0.377 for depression.

*A moderated analysis of COVID-19 stress on anxiety: education level.* Anxiety levels were lower in graduate international students than in undergraduate international students as indicated by the significant (and negative) coefficient for education (β = −0.11, p < 0.05; Table 3). Our hypothesis that COVID-19 stress was associated with an increase in anxiety was supported by a significant relationship between COVID-19 stress and anxiety (β = 0.47, p < 0.01). Therefore, for every 1 SD increase in COVID-19 stress, anxiety scores increased by 0.47 SD units. The

**Table 2. Pearson correlation coefficients among psychological variables used in the model.**

|  | Global Stress | COVID-19 Stress | Anxiety | Depression | Age | Sex | Marital Status |
|---|---|---|---|---|---|---|---|
| COVID-19 Stress | 0.28** |  |  |  |  |  |  |
| Anxiety | 0.46** | 0.46** |  |  |  |  |  |
| Depression | 0.58** | 0.34** | 0.60** |  |  |  |  |
| Age | −0.14* | −0.11 | −0.11 | −0.11 |  |  |  |
| Sex | 0.11 | 0.09 | 0.06 | −0.01 | −0.09 |  |  |
| Marital Status | −0.19** | 0.01 | −0.06 | −0.13* | 0.42** | −0.19** |  |
| Education Level | −0.17* | −0.12 | −0.21** | −0.11 | 0.52** | −0.23** | 0.19** |

*p < 0.05, **p < 0.01.

**Table 3. Direct and indirect effects for the model.**

| Outcome Variable | Predictors | Unstandardized coefficient (B) | Standardized coefficient (β) | R-Square |
|---|---|---|---|---|
| Anxiety | Intercept | 4.280** |  | 0.352 |
|  | Global Stress | 0.486** | 0.346** |  |
|  | COVID-19 Stress | 0.805** | 0.471** |  |
|  | Education level | −0.899* | −0.109* |  |
|  | Interaction term[a] | −0.480* | −0.191* |  |
| Depression | Intercept | 6.262** |  | 0.377 |
|  | Global Stress | 1.078** | 0.519** |  |
|  | COVID-19 Stress | 0.510** | 0.202** |  |
|  | Education level | 0.153 | 0.013 |  |
|  | Marital status | −1.043 | −0.053 |  |
|  | Interaction term[a] | −0.015 | −0.004 |  |

[a] COVID-19 stress x Education level.

*p < 0.05, **p < 0.01.

hypothesis that education level moderates the relationship between COVID-19 stress and anxiety was supported by the significant coefficient of the interaction term ($\beta = -0.19$, $p < 0.05$). This means that the strength of the relation between COVID-19 stress and anxiety was stronger in undergraduates ($\beta = 0.47$) than in graduates ($\beta = 0.28$). Therefore, being an undergraduate increased the effect of COVID-19 stress and anxiety.

***A moderated analysis of COVID-19 stress on depression: Education level.*** We found no differences in depression levels among undergraduate and graduate international students ($\beta = 0.01$, $p > 0.05$). The hypothesis that COVID-19 stress was associated with an increase in depression was supported by a significant relationship between COVID-19 stress and depression ($\beta = 0.20$, $p < 0.01$). Therefore, for every 1 SD increase in COVID-19 stress, depression scores increased by 0.20 SD units. Additionally, we hypothesized that education level moderates the relationship between COVID-19 stress and depression. However, this hypothesis was not supported ($\beta = -0.004$, $p > 0.05$).

## COVID-19-related stressors affecting the lives of international students

With regards to specific stressors that were affecting the mental health of international students during the COVID-19 pandemic, we found the following themes: the stresses of academic performance, financial difficulties, travel constraints, social isolation, and dealing with the loss of loved ones and grief.

*Academic Performance*. The effect on academic performance, such as adjusting to remote learning, focusing, meeting the demands of course requirements, and maintaining grades were the most reported stressor for international students, *"Yes, covid has impacted my ability to concentrate on my courses and my grades have gone down because of it. I have been trying hard…meanwhile the whole world is upside down for people..."*—25–30 years, male, single, graduate. Some students even felt that faculty were not empathetic or supportive, *"It has definitely made the learning process much more difficult because it's very harder to absorb the content when compared to in-person classes. Also, you're constantly bombarded by assignments, homework and [Microsoft] Teams meetings and it seems that the teachers are disconnected with reality and with the fact that we are really struggling trying to keep up. They don't facilitate the learning process. They believe due to the fact that we are "at home", we function at the same level than any other semester, but they are wrong. They need to be more considerate and help us out more."* —25–30 years, male, married, undergraduate.

*Financial Difficulties*. International students experienced financial difficulties due to lack of institution funding and scholarships, on-campus job loss due to school closure, *"No help from the university as a DACA student, paying full-time tuition, no payment deferment, no financial help. Killed myself providing for my family since they are jobless due to COVID and trying to continue my education through this institution that does not help me in the slightest."* —18–24 years, female, single, undergraduate, and loss of financial support from family. Some students were burdened with the added responsibility of financially supporting their family in their home country because of job loss or illness from COVID-19 infection to immediate family members, *"The covid-19 has affected me and my family financially. I have been experiencing a lot of financial difficulties such as rent, food and school supplies."* —18–24 years, female, single, undergraduate; *"My family got sick with Covid in India, so I had to send them money which put me in a financial burden. I'm still struggling with that."* —25–30 years, female, single, graduate.

*Traveling Constraints.* During the pandemic, the borders of many countries globally including the US were closed, and new visa restrictions were imposed for international students studying at US colleges and universities. Many international students feared that they were being forcefully expelled from the US and could not complete their studies, *"Just worrying about my family in general and feeling scared that I will not be able to see them in a long time. Also, it is hard to think about what the government might do with us since we have had two attempts from the government to kick us out. Lastly, I am worried about not being able to finish up my studies because of Covid and also not getting my visa renewed."* —18–24 years, female, single, undergraduate; *"Yes, I had to go back to my home country to stay with my family in case something happened since I live by myself in the US. Now that I live outside the US…I had to pay an extra flight ticket which was not planned at all and left my belongings there because of the last-minute emergency. Since today, I still pay*

_rent, car insurance, and medical insurance even though I do not live there and I cannot go back to bring back my belongings because the US borders remain closed."_ —18–24 years, female, single, graduate. For some international students, the added stressor of traveling constraints manifested as physical, emotional, and psychological health symptoms, _"In summer, when ICE announced a new policy about F1 students (who will have to return home in a few weeks if they don't have in-person classes in Fall) - I was so nervous that I got an awful skin disease due to stress. It wasn't an infection or an allergy - just a stress-provoked disease. Since March, I live with the heaviest depression and bad thoughts about my life. I have constant headaches and panic attacks at night, sleep disorder, and loss of concentration. My mind is not working well anymore. And now, when I can't get my internship, which was first confirmed one month ago and then - silence, I start panicking again and can't recover from it. I paid for internship my last money and receive nothing but frustration."_ —35+years, female, married, graduate.

_Social Isolation_. As expected, many students reported struggling with the inability to socialize or engage in social activities because of lockdown and social distancing measures, leaving them feeling socially isolated and lonely which further had an effect on their mental health, _"I have not been able to see my friends much since they are also doing college online and for safety measures. This leaves me at home alone a lot of times with my thoughts and it becomes very overwhelming and dark."_ —18–24 years, female, single, undergraduate; _"Not been going to class, I miss my classmates, study group sessions, and meeting people in club activities, meeting people online just is not the same."_ —18–24 years, male, single, graduate.

_Death and Grief_. International students also reported having to deal with the death of loved ones, their own grief, and the grief of other family members because of the pandemic, _"Several people I knew died from covid-19: a colleague from high school (fem, 27 yrs), family friend sort of like an uncle (male, 38 yrs), a member of my Buddhist group's husband very much like an uncle (male, 52 yrs), streaming community member's brother (male, 49 yrs). These deaths occurred both within the United States (Wisconsin, Florida) and outside of the United States (St. Lucia, Trinidad and Tobago). They heavily impacted friends and family alike, as a result I was impacted too because their grief is my grief, shouldering loss is difficult, we only have each other, and there's so many other factors dividing everyone when we really need to just work together for everyone's sake."_ —25–30 years, female, married, undergraduate.

A joint display (Table 4) is used to present integration of the mixed-methods findings of key quantitative associations with corresponding qualitative themes and participant quotes.

## Discussion

This study shed light on an important but under-investigated topic in the COVID-19 pandemic, which is the identification of a negative impact on the mental health of international students. COVID-19 stress, anxiety, and depression were major global issues that affected all students, but our findings indicate specific effects of COVID-19 stress on anxiety and depressive symptoms among international students.

Our study is among the very few college health studies that have focused on international students during the COVID-19 pandemic in Florida [5,45,46] and contributed with an examination of how COVID-19 specific stress was related to self-reported anxiety and depression among international students attending a US higher education institution. We also found that COVID-19 stress interacted with education level (undergraduate vs graduate level) on its effects on anxiety, indicating that undergraduate students were stressed due to COVID, but this resulted in higher levels of anxiety compared to the anxiety level reported by graduate students ($\beta = -0.19$). This suggests that undergraduate students were more vulnerable to develop anxiety from being exposed to COVID-19-related stress evidenced by qualitative narratives of undergraduate students feeling overwhelmed by the demands of remote learning and their perceptions of inadequate faculty support, with one student sharing that _"…teachers are disconnected with reality…we are really struggling"_, and experiencing social isolation from friends, where another student shared, _"I have not been able to see my friends…and it becomes very overwhelming and dark"_. These findings were similar to those found by Wang et al [7], where undergraduate

**Table 4. Integrated Joint Display of Quantitative and Qualitative Findings on COVID-19 Stress, Anxiety, and Depression in International Students.**

| Quantitative Findings | Themes | Participant Quotes |
|---|---|---|
| COVID-19 stress was significantly associated with **anxiety**, with a stronger effect for undergraduates ($\beta = 0.471$, interaction $\beta = -0.191$). | The pressures of **academic performance** and perceived lack of support from faculty, heightened anxiety, particularly among undergraduates. | "It has definitely made the learning process much more difficult…teachers are disconnected with reality…They need to be more considerate and help us out more." —25–30 yrs, male, married, undergraduate. |
| Global stress was significantly associated with **anxiety** ($\beta = 0.346$). | Emotional distress was increased by **social isolation**, overwhelming students. | "I have not been able to see my friends much…This leaves me at home alone a lot with my thoughts and it becomes very overwhelming and dark." —18–24 yrs, female, single, undergraduate |
| COVID-19 stress was significantly associated with **depression** across all students ($\beta = 0.202$), with no moderation by education level. | **Travel constraints** along with **death and grief** were prevalent sources of distress. | "In summer, when ICE announced a new policy about F1 students who will have to return home in a few weeks…I live with the heaviest depression and bad thoughts about my life…My mind is not working well anymore." —35+ years, female, married, graduate. "Several people I knew died from covid-19…their grief is my grief, shouldering loss is difficult..." —25–30 yrs, female, married, undergraduate |
| Global stress was significantly associated with **depression** ($\beta = 0.519$). | **Financial difficulties** added to students' stress and uncertainty. | "My family got sick with Covid in India, so I had to send them money which put me in a financial burden. I'm still struggling with that." —25–30 yrs, female, single, graduate "I have been experiencing a lot of financial difficulties such as rent, food and school supplies." —18–24 years, female, single, undergraduate |

students reported more anxiety symptoms than did graduates. Liu et al [22] also found that US undergraduate students experienced poorer mental health due to the influence of pandemic-related stress, but these authors did not differentiate between specific mental health conditions. When our survey was conducted, there was no access to vaccines and the situation was rapidly changing with a relentless virus continuing to take its death toll in the US and globally [47]. Indeed, the impact of a major global infectious emergency, as it was the COVID-19 pandemic, likely generated very high levels of uncertainty among international students. Perceived uncertainty about a the COVID-19 pandemic may have impaired the students' ability to lessen its negative impact, and therefore resulted in anxiety [48]. Regarding the differences in anxiety between undergraduate and graduate students, there is evidence of greater social support among graduate students in general [49]. Yet, studies examining this phenomenon among international students are lacking. However, international students often migrate with their families, a well-known source of social support. Lastly, we speculate that there may be differences regarding experiences handling stress by graduate students, levels of maturity, and more developed coping skills among graduate students. However, these factors were not measured in our study. We recommend future studies are conducted to assess coping mechanisms among graduate and undergraduate international students.

Although Wang et al [50] also found significant differences on depression by education level, we did not find a significant interaction. Specifically, we found that depression was predicted by COVID-19 stress for all international students, but there was no significant interaction with education level. It is possible that depression among international students may be influenced by universal and uncontrollable stressors such as separation from family and the inability to travel home, family illness, death and grief, and extended uncertainty about the pandemic, rather than by pressures related to students' academic level [51,52]. For instance, one student expressed *"…I live with the heaviest depression and bad thoughts about my life…"* as a result of the travel constraints imposed, with another sharing *"Several people I knew died from COVID-19…their grief is my grief"*, highlighting how travel constraints and death and grief heightened psychological distress over educational stressors. Financial burden was a major contributor described mostly due to job loss and the additional stress of supporting family abroad, where one student said, *"My family got sick with Covid in India, so I had to send them money which put me in a financial burden."* These qualitative responses contextualize the significant

associations between COVID-19 stress, global stress, and depression across education levels. Furthermore, it should be noted that the CSS focused on general stressors unrelated to academia such as fears of contracting COVID-19 infection, socioeconomic concerns, COVID-19 traumatic stress symptoms, and compulsive checking behaviors, which are more likely to affect students uniformly irrespective of their education level, explaining similarities in depression. Browning et al [53] also found no differences on the psychological impact (combined measure: negative emotion states, preoccupation with COVID-19, feeling stressed, worry, and time demands) of the pandemic on undergraduate students compared to graduates. We speculate that late 2020 (during the time of our survey), which was a period marked by greater uncertainty and significant changes in the US (e.g., elections, lack of uniform protections), influenced depression symptoms, but not necessarily hopelessness. As the pandemic prolonged for another year, we would have expected to see more feelings of hopelessness, grief (due to personal and community loss), and isolation in 2021.

These studies and ours were conducted in single institutions and distinct geographical settings where conditions and access to mental health support in the community may vary for students. As more studies examine the differential impact of the pandemic on students' mental health by undergraduate/graduate status, it is necessary that meta-analyses are conducted to generate pooled estimates accounting for differences in sample sizes and other conditions, such as access to mental health services.

The open-ended question about ways that the COVID-19 pandemic impacted the students' academics, health, and life overall revealed that both international undergraduate and graduate students encountered numerous stressors that have had an effect on their mental health. The list of stressors included worries about academic performance, financial difficulties, travel constraints, socialization difficulties, and dealing with the loss of loved ones and grief. With regards to academic issues, Zhang et al [54] reported that international students faced limited understanding by educators and staff regarding their unique situation. Based on our findings, we recommend that higher education institutions should monitor the psychological well-being of international students more actively and implement support mechanisms to improve the academic performance among international students.

We consider that financial difficulties were likely exacerbated as international students are constrained about their employment permits, which depends on visa status. For instance, the majority of international students hold student visas that allow them to work in university settings only. There were reports of significant obstacles to conduct university research during the pandemic, which is a major source of international students' work. This may have impacted on job availability at the university. The inability to travel back home for international students likely contributed to increased anxiety and depression as they witnessed their relatives in their home countries where COVID-19 was exerting major death tolls as healthcare systems with fewer resources couldn't compensate the demands. Many suffered deaths in their families, which resulted in personal grief. Due to travel constraints, they were not able or allowed to attend family funerals or were unable to receive or bring help to their loved ones. Socialization difficulties and isolation were also reported by other authors [54].

The quantitative strand has inherent limitations related to the cross-sectional nature of the data, for which causality cannot be inferred. Additionally, confounders which were not examined such as prior mental health history, coping style, visa status, social support, and cultural distance may influence anxiety and depression, limiting causal interpretation. The relatively small sample, the single institution setting in Florida, and timing during the early phase of the pandemic prior to the COVID-19 vaccine, are also limitations that limit the external validity of the study. Therefore, study findings may not be generalized to other colleges/universities, later phases of the pandemic, or countries with different policies or support systems. The fact that we were not able to provide incentives limited our ability to increase survey responses. We recommend additional studies be conducted to confirm our findings with more representative samples, using probabilistic sampling.

The qualitative strand also has limitations inherent to online qualitative surveys, including brevity of responses and no personalized follow-up questions. The number of responses, however, facilitated the attainment of saturation for the

identified themes. We recommend future qualitative studies be conducted using in-person techniques to take a deeper dive on the themes we identified. Because of the self-reporting nature of mental health scales, social desirability and recall bias may have influenced participant responses. However, the anonymous online administration of the survey may reduce these biases but not eliminate them. Another limitation is that we did not assess cultural differences among international students. Cultural group membership has been found to be a significant predictor of coping mechanisms that predict psychological well-being. We recommend future studies are conducted with cultural sub-group analyses to identify cultural differences.

Overall, our voluntary recruitment through email and social media may have introduced selection bias, as students who were more engaged or affected by the pandemic were more likely to have participated than those who were less connected or affected, thereby limiting the generalizability of study findings.

International students continue to experience unique challenges. As we have transitioned into a post-COVID "new normal", colleges and universities should pay greater attention to international students, particularly undergraduates, during difficult times such as public health emergencies. Such efforts must also be supported by work or travel arrangements that allow international students to meet their financial needs during times of crisis. Universities can also have a key role to ameliorate anxiety-producing perceptions of students by providing additional supports coming from instructors, university administration, and student networks [55].

## Conclusion

COVID-19 stress increased anxiety in international students, particularly undergraduates, and depression across both international undergraduates and graduates. Underlying mechanisms of these associations were identified as academic pressure, financial strain, travel constraints, social isolation, and grief from qualitative responses. Therefore, it is imperative that universities provide targeted support to international students to ameliorate the effects of stress on students' mental health, including during future crises.

## Supporting information

**S1 Table. Item-Total Correlations and Cronbach's Alpha for the Modified COVID-19 Stress Scale.**
(DOCX)

**S1 Data. Dataset and Code.**
(ZIP)

**S2 Data. International Students COVID-19 Dataset.**
(ZIP)

## Acknowledgments

We would like to thank the international students who participated in this study and Marcia Taylor and Kristen Zernick of the Office of International Services, USF World.

## Author contributions

**Conceptualization:** Chinyere N. Reid.

**Data curation:** Chinyere N. Reid.

**Formal analysis:** Chinyere N. Reid, Jason W. Beckstead.

**Investigation:** Chinyere N. Reid.

**Methodology:** Chinyere N. Reid, Jason W. Beckstead.

**Project administration:** Chinyere N. Reid.

**Supervision:** Jason W. Beckstead, Abraham A. Salinas-Miranda.

**Validation:** Jason W. Beckstead, Abraham A. Salinas-Miranda.

**Visualization:** Chinyere N. Reid.

**Writing – original draft:** Chinyere N. Reid, Abraham A. Salinas-Miranda.

**Writing – review & editing:** Chinyere N. Reid, Jason W. Beckstead, Abraham A. Salinas-Miranda.

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
