## [Decision Letter · Decision Letter 0]

26 Jun 2025

Dear Dr. Reid,

Thank you for submitting your manuscript to PLOS ONE. After careful consideration, we feel that it has merit but does not fully meet PLOS ONE’s publication criteria as it currently stands. Therefore, we invite you to submit a revised version of the manuscript that addresses the points raised during the review process.

We look forward to receiving your revised manuscript.

Kind regards,

Mukhtiar Baig, Ph.D.

Academic Editor

PLOS ONE

2. In the online submission form, you indicated that [The data underlying the results presented in the study are available from the corresponding author.].

Additional Editor Comments (if provided):

Reviewers' comments:

Reviewer's Responses to Questions

**Comments to the Author**

1. Is the manuscript technically sound, and do the data support the conclusions?

Reviewer #1: Yes

Reviewer #2: Partly

2. Has the statistical analysis been performed appropriately and rigorously?

Reviewer #1: Yes

Reviewer #2: Yes

3. Have the authors made all data underlying the findings in their manuscript fully available?

Reviewer #1: Yes

Reviewer #2: No

4. Is the manuscript presented in an intelligible fashion and written in standard English?

Reviewer #1: Yes

Reviewer #2: Yes

Reviewer #1: The paper accurately reflects the focus of the study—exploring the relationship between COVID-19-related stress and mental health outcomes in international students. The use of a mixed-methods approach is also clearly signaled, which is a strength, as it suggests both quantitative and qualitative data are utilized.

This paper is a valuable contribution to the growing body of research on the pandemic’s mental health impact. It particularly underscores the unique stressors international students face. While some methodological clarifications may be needed, the paper offers important insights and practical implications for improving mental health support systems in higher education.

Reviewer #2: 1. The voluntary email- and social-media recruitment strategy is likely to have attracted students who were already engaged or distressed; please acknowledge this selection bias and explain how it limits generalizability to the wider international‐student population—especially those who are less connected to university listservs or who disengaged during the pandemic.

2. The study is single-institution and Florida-specific, with data collected in October 2020 (pre-vaccine, high policy uncertainty). Add a brief paragraph clarifying how time, place, and institutional context constrain external validity; readers need to know that results may not transfer to other campuses, later stages of the pandemic, or countries with different support systems.

3. Provide basic regional or cultural origin data (e.g., East Asia, Latin America, South Asia). If they were not collected, state this explicitly as a limitation because pandemic experiences and acculturative stress vary by region.

4. The xenophobia subscale of the COVID-Stress Scale was removed, yet discrimination is central to international-student stress. Offer a detailed rationale (e.g., conceptual mismatch, redundancy, statistical misfit), report any reliability or validity analyses you ran with and without that subscale, and discuss what information may have been lost by excluding it.

5. Explain why path analysis was chosen instead of a simpler multiple-regression framework. If the intent was to model simultaneous outcomes, indirect effects, or moderation, state that plainly and note any underlying assumptions (e.g., multivariate normality, linearity) and diagnostics performed.

6. Clarify whether the path coefficients were standardized using the rescaled CSS variable (divided by 10). Readers will want to know if interpretation of effect sizes aligns across differently scaled measures.

7. Unmeasured confounders—prior mental-health history, coping style, visa status, social support, cultural distance—could affect both stress and mental-health outcomes. A short paragraph in the limitations acknowledging these unrecorded factors would temper causal claims.

8. Missing data were minimal, but specify how you verified that data were missing completely at random before proceeding with listwise deletion. If you ran Little’s MCAR test or examined patterns, report it.

9. Adopt a joint display to integrate quantitative and qualitative findings. A simple matrix could list each significant quantitative association (e.g., COVID-19 stress → anxiety stronger for undergraduates) next to illustrative quotes and thematic interpretations (academic workload, faculty expectations). This single table or figure would satisfy mixed-methods best practice and make triangulation evident.

10. Integration in the discussion is currently light. Deliberately weave qualitative insights into each key quantitative result—e.g., link the regression coefficient for financial strain to the narrative of job loss and remittances; use grief quotes to contextualize the depression path coefficient.

11. The qualitative strand relies on a single open-ended question and one coder. Describe how you ensured rigor—inter-coder reliability (even a second coder on a 20 % subsample), peer debriefing, reflexive journaling, or member checking. If none were applied, state that explicitly and recognize the interpretive limitation.

12. Applied thematic analysis usually involves iterative coding, codebook refinement, and theme saturation checks. Outline the exact steps: how many iterations, how code definitions evolved, how you decided saturation was reached with 115 responses, and whether you tracked theme-frequency counts.

13. Consider adding sub-theme counts or proportions (e.g., “40 % of undergraduates mentioned academic pressure vs 25 % of graduates”). Even simple counts in the joint display would help readers judge the weight of each theme across groups.

14. Offer a brief reflexivity statement: describe the coder’s positionality (international student? domestic? researcher role?) and how that may have influenced interpretation of narrative responses.

15. The discussion of the non-significant moderation for depression is brief. Speculate why education level does not modify this pathway—perhaps depressive symptoms arise from universal stressors (loss, isolation) less tied to academic role—and reference any supporting literature.

16. Provide clearer reasoning for scaling the CSS scores by dividing by 10. While it aids comparability, reassure readers that scaling did not distort reliability or model estimates, and note that scaling does not affect significance.

17. The Cronbach’s alpha values are strong, but add item-total or subscale alphas in an online supplement; this would be especially helpful given the CSS modification.

18. Acknowledge social-desirability and recall bias inherent in self-reported mental-health scales, and note that online anonymity may mitigate but not eliminate these effects.

19. Table 1 and Table 2 contain a mix of continuous means and ordinal age categories; consider reporting median and IQR for skewed variables or justify why means are appropriate despite large SDs (e.g., anxiety).

20. You reference “normality assumptions were met” for path analysis; briefly state how this was checked (e.g., skewness/kurtosis thresholds, inspection of histograms).

21. The IRB description is inconsistent—one section is blinded, another lists the board and approval ID. Align them (non-blinded) to avoid reader confusion and to meet PLOS data-sharing transparency norms.

22. End with a concise conclusion section that recaps the integrated findings: “COVID-19–specific stress increased anxiety (especially for undergraduates) and depression in all international students; qualitative narratives identify academic pressure, financial strain, travel constraints, social isolation, and grief as underlying mechanisms. Universities should…” This will give readers a clear takeaway.

23. Finally, recommend posting the anonymized dataset and analytic code (Mplus and Excel coding files) in a public repository to adhere to PLOS ONE’s data-availability standards and enhance reproducibility.

**Do you want your identity to be public for this peer review?** For information about this choice, including consent withdrawal, please see our Privacy Policy

Reviewer #1: **Yes:** Milan Latas

Reviewer #2: No

---

## [Author Response · Author response to Decision Letter 1]

1 Oct 2025

We have attached our "Response to Reviewers' letter" and also copy/pasted here for convenience:

Reviewer 1, Comment 1:

The paper accurately reflects the focus of the study—exploring the relationship between COVID-19-related stress and mental health outcomes in international students. The use of a mixed-methods approach is also clearly signaled, which is a strength, as it suggests both quantitative and qualitative data are utilized.

This paper is a valuable contribution to the growing body of research on the pandemic’s mental health impact. It particularly underscores the unique stressors international students face. While some methodological clarifications may be needed, the paper offers important insights and practical implications for improving mental health support systems in higher education.

Response:

Thank you for the positive feedback.

Reviewer 2, Comment 1:

The voluntary email- and social-media recruitment strategy is likely to have attracted students who were already engaged or distressed; please acknowledge this selection bias and explain how it limits generalizability to the wider international‐student population—especially those who are less connected to university listservs or who disengaged during the pandemic.

Response: Thank you for highlighting this. We agree with this observation and have added text to the Limitations section to acknowledge this potential selection bias and its impact on generalizability. See Lines 461-464

Reviewer 2, Comment 2:

The study is single-institution and Florida-specific, with data collected in October 2020 (pre-vaccine, high policy uncertainty). Add a brief paragraph clarifying how time, place, and institutional context constrain external validity; readers need to know that results may not transfer to other campuses, later stages of the pandemic, or countries with different support systems.

Response: Thank you for your comment. We previously mentioned “The relatively small sample and the single institution setting are two other limitations, which are common in college student surveys and limit the external validity of the study.” in the Limitations section. We have expanded this to include how the study’s timing and pre-vaccine context may limit generalizability. We now note that findings may not transfer to other institutions, stages of the pandemic, or international settings with different support systems. Lines 442-446

Reviewer 2, Comment 3:

Provide basic regional or cultural origin data (e.g., East Asia, Latin America, South Asia). If they were not collected, state this explicitly as a limitation because pandemic experiences and acculturative stress vary by region.

Response: Thank you for this suggestion. We have added students’ regions of origin to the Results section. Lines 236-239

Reviewer 2, Comment 4:

The xenophobia subscale of the COVID-Stress Scale was removed, yet discrimination is central to international-student stress. Offer a detailed rationale (e.g., conceptual mismatch, redundancy, statistical misfit), report any reliability or validity analyses you ran with and without that subscale, and discuss what information may have been lost by excluding it.

Response: The xenophobia subscale of the COVID Stress Scale (CSS) measures xenophobic attitudes held by participants toward others (e.g., fear or mistrust of foreigners), which posed a conceptual mismatch in the context of our international student sample—who are more likely to be targets rather than sources of such attitudes. Including this subscale could have been inappropriate or confusing for participants. We opted not to modify the items, as altering a validated scale could compromise measurement validity. No reliability or validity analyses were run with the xenophobia subscale, as it was excluded a priori. We acknowledge the potential loss of insight into participants’ experiences of being targeted by xenophobia. Lines 139-144

Reviewer 2, Comment 5:

Explain why path analysis was chosen instead of a simpler multiple-regression framework. If the intent was to model simultaneous outcomes, indirect effects, or moderation, state that plainly and note any underlying assumptions (e.g., multivariate normality, linearity) and diagnostics performed.

Response: We chose path analysis over multiple regression to test a hypothesized model that included both direct effects and moderation, while accounting for interrelationships among multiple dependent variables (anxiety, and depression). Path analysis allows for simultaneous estimation of multiple outcome paths and the inclusion of interaction terms within a single structural framework. We used Maximum Likelihood Estimation (MLE) in path analysis, which is more versatile and robust than ordinary least squares (OLS) regression (MLR). MLE handles complex model structures more effectively and is more robust to violations of assumptions (e.g., non-normality) under larger samples. Chambers, R. L. (2012). Maximum Likelihood Estimation for Sample Surveys. CRC Press. https://doi.org/10.1201/b12038

We have also clarified that distributional properties were assessed using skewness, kurtosis, and histogram inspection to ensure appropriateness for maximum likelihood estimation (MLE). Lines 192-197

Reviewer 2, Comment 6:

Clarify whether the path coefficients were standardized using the rescaled CSS variable (divided by 10). Readers will want to know if interpretation of effect sizes aligns across differently scaled measures.

Reponse: Yes, the path coefficients were standardized using the rescaled CSS variable (divided by 10) to improve interpretability. Standardized results were presented in Table 3.

Reviewer 2, Comment 7:

Unmeasured confounders—prior mental-health history, coping style, visa status, social support, cultural distance—could affect both stress and mental-health outcomes. A short paragraph in the limitations acknowledging these unrecorded factors would temper causal claims.

Response: We recognize that other factors such as prior mental-health history, coping style, visa status, social support, cultural distance can affect mental health, and have expanded the limitations section to include this. Lines 440-442

Reviewer 2, Comment 8:

Missing data were minimal, but specify how you verified that data were missing completely at random before proceeding with listwise deletion. If you ran Little’s MCAR test or examined patterns, report it.

Response: Because missingness was less than 5% and Little’s MCAR test (p = 0.078) indicated that the data was missing completely at random, listwise deletion was deemed appropriate. Lines 186-188

Reviewer 2, Comment 9:

Adopt a joint display to integrate quantitative and qualitative findings. A simple matrix could list each significant quantitative association (e.g., COVID-19 stress → anxiety stronger for undergraduates) next to illustrative quotes and thematic interpretations (academic workload, faculty expectations). This single table or figure would satisfy mixed-methods best practice and make triangulation evident.

Response: Thank you for this suggestion. To enhance triangulation, we have added a joint display table integrating key quantitative findings with supporting qualitative themes and illustrative quotes. Lines 344-345; Table 4

Reviewer 2, Comment 10:

Integration in the discussion is currently light. Deliberately weave qualitative insights into each key quantitative result—e.g., link the regression coefficient for financial strain to the narrative of job loss and remittances; use grief quotes to contextualize the depression path coefficient.

Response: We revised the Discussion to explicitly integrate qualitative findings with each key quantitative finding to contextualize the observed statistical associations. Lines 360-366, 387-399

Reviewer 2, Comment 11:

The qualitative strand relies on a single open-ended question and one coder. Describe how you ensured rigor—inter-coder reliability (even a second coder on a 20 % subsample), peer debriefing, reflexive journaling, or member checking. If none were applied, state that explicitly and recognize the interpretive limitation.

Response: While coding was conducted by a single researcher, we ensured rigor through peer debriefing, which involved regular discussions with another researcher to critically review coding decisions and interpretations. This process enhanced credibility and reduced potential interpretive bias. Lines 220-222

Reviewer 2, Comment 12:

Applied thematic analysis usually involves iterative coding, codebook refinement, and theme saturation checks. Outline the exact steps: how many iterations, how code definitions evolved, how you decided saturation was reached with 115 responses, and whether you tracked theme-frequency counts.

Response: We have added to the Methods section to describe the iterative coding process (two coding cycles), codebook refinement, and how data saturation was determined (no new themes emerging after reviewing all 115 responses). We did not track theme frequency—described below. Lines 217-218, 222-223

Reviewer 2, Comment 13:

Consider adding sub-theme counts or proportions (e.g., “40 % of undergraduates mentioned academic pressure vs 25 % of graduates”). Even simple counts in the joint display would help readers judge the weight of each theme across groups.

Response: We did not quantify the qualitative data because our analysis followed a reflexive thematic approach in which themes are not determined by frequency but by their relevance to the research question. Quantifying themes can be misleading in this context and is not a requirement of thematic analysis (Braun & Clarke, 2006).

Braun, V., & Clarke, V. (2006). Using

thematicanalysis in psychology.

Qualitative Research in Psychology, 3(2),

77–101.

Reviewer 2, Comment 14:

Offer a brief reflexivity statement: describe the coder’s positionality (international student? domestic? researcher role?) and how that may have influenced interpretation of narrative responses.

Response: We added a reflexivity statement to the Methods section. We acknowledged that the coder was an international graduate student researcher and that this positionality may have introduced interpretive bias. However, peer debriefing was conducted to mitigate this. Lines 218-222

Reviewer 2, Comment 15:

The discussion of the non-significant moderation for depression is brief. Speculate why education level does not modify this pathway—perhaps depressive symptoms arise from universal stressors (loss, isolation) less tied to academic role—and reference any supporting literature.

Response: We have expanded the Discussion to speculate that depressive symptoms may reflect universal, uncontrollable stressors (e.g., grief, uncertainty, family separation) rather than academic‑role–specific pressures. We also noted that the CSS primarily captures stress domains unrelated to academia, which may explain similar depressive responses across education levels. Lines 387-402

Reviewer 2, Comment 16:

Provide clearer reasoning for scaling the CSS scores by dividing by 10. While it aids comparability, reassure readers that scaling did not distort reliability or model estimates, and note that scaling does not affect significance.

Response: We divided the CSS scores by 10 to improve interpretability of path coefficients and facilitate comparability with other variables. This linear transformation does not affect reliability, significance, or the relative magnitude of estimates, as scaling preserves variance and model fit. Lines 180-182

Reviewer 2, Comment 17:

The Cronbach’s alpha values are strong, but add item-total or subscale alphas in an online supplement; this would be especially helpful given the CSS modification.

Response: While the overall Cronbach’s alpha values were strong and the CSS modification was carefully justified, we agree that providing item-total correlations and subscale alpha values will enhance transparency. We have included these psychometric details for the modified CSS in the supplement. Lines 148-149

Supplementary Table 1

Reviewer 2, Comment 18:

Acknowledge social-desirability and recall bias inherent in self-reported mental-health scales, and note that online anonymity may mitigate but not eliminate these effects.

Response: We have added to the Limitations section acknowledging that self-reported mental health data are subject to social-desirability and recall bias, which may influence responses. While online anonymous administration may mitigate these biases, it does not eliminate them. Lines 440-442

Reviewer 2, Comment 19:

Table 1 and Table 2 contain a mix of continuous means and ordinal age categories; consider reporting median and IQR for skewed variables or justify why means are appropriate despite large SDs (e.g., anxiety).

Response: Thank you for this suggestion. We have updated Table 1 to report median and interquartile range. See Table 1

Reviewer 2, Comment 20:

You reference “normality assumptions were met” for path analysis; briefly state how this was checked (e.g., skewness/kurtosis thresholds, inspection of histograms).

Response: We have clarified that distributional properties were assessed using skewness, kurtosis, and histogram inspection to ensure appropriateness for maximum likelihood estimation (MLE). Lines 196-197

Reviewer 2, Comment 21:

The IRB description is inconsistent—one section is blinded, another lists the board and approval ID. Align them (non-blinded) to avoid reader confusion and to meet PLOS data-sharing transparency norms.

Response: We initially used “[BLINDED University]” as a placeholder for the peer-review process, but we have now removed this placeholder and stated the university name. Line 229

Reviewer 2, Comment 22:

End with a concise conclusion section that recaps the integrated findings: “COVID-19–specific stress increased anxiety (especially for undergraduates) and depression in all international students; qualitative narratives identify academic pressure, financial strain, travel constraints, social isolation, and grief as underlying mechanisms. Universities should…” This will give readers a clear takeaway.

Response: Thank you for this suggestion. We have added a concise Conclusion section that integrates the key quantitative and qualitative findings and provides a clear takeaway. Lines 473-478

Reviewer 2, Comment 23:

Finally, recommend posting the anonymized dataset and analytic code (Mplus and Excel coding files) in a public repository to adhere to PLOS ONE’s data-availability standards and enhance reproducibility.

Response: Thank you for the recommendation. We have posted the anonymized dataset in a public repository, in accordance with PLOS ONE’s requirements.

---

## [Decision Letter · Decision Letter 1]

26 Oct 2025

Exploring the influence of COVID-19 stress on mental health among international undergraduate and graduate students: A mixed-methods approach

PONE-D-25-25688R1

Dear Dr. Salinas-Miranda,

We’re pleased to inform you that your manuscript has been judged scientifically suitable for publication and will be formally accepted for publication once it meets all outstanding technical requirements.

Kind regards,

Mukhtiar Baig, Ph.D.

Academic Editor

PLOS ONE

Reviewers' comments:

Reviewer's Responses to Questions

**Comments to the Author**

Reviewer #2: All comments have been addressed

2. Is the manuscript technically sound, and do the data support the conclusions?

Reviewer #2: Yes

3. Has the statistical analysis been performed appropriately and rigorously?

Reviewer #2: Yes

4. Have the authors made all data underlying the findings in their manuscript fully available?

Reviewer #2: No

5. Is the manuscript presented in an intelligible fashion and written in standard English?

Reviewer #2: Yes

Reviewer #2: All reviewer suggestions have been appropriately incorporated or justified, with line references, new text additions, and supplementary materials confirming compliance with PLOS ONE’s methodological and data-sharing standards.

**Do you want your identity to be public for this peer review?** For information about this choice, including consent withdrawal, please see our Privacy Policy

Reviewer #2: No

---

## [Editor Report · Acceptance letter]

PONE-D-25-25688R1

PLOS One

Dear Dr. Salinas-Miranda,

I'm pleased to inform you that your manuscript has been deemed suitable for publication in PLOS One. Congratulations! Your manuscript is now being handed over to our production team.

Kind regards,

on behalf of

Professor Mukhtiar Baig

Academic Editor

PLOS One